# Vitrification-enabled enhancement of proton conductivity in hydrogen-bonded organic frameworks

Feng-Fan Yang [1], Xiao-Lu Wang[1,2], Jiayue Tian[3], Yang Yin[1] & Linfeng Liang [1] ✉

Hydrogen-bonded organic frameworks (HOFs) are versatile materials with potential applications in proton conduction. Traditional approaches involve incorporating humidity control to address grain boundary challenges for proton conduction. This study finds vitrification as an alternative strategy to eliminate grain boundary effect in HOFs by rapidly melt quenching the kinetically stable **HOF-SXU-8** to glassy state **HOF-g**. Notably, a remarkable enhancement in proton conductivity without humidity was achieved after vitrification, from $1.31 \times 10^{-7}\,S\,cm^{-1}$ to $5.62 \times 10^{-2}\,S\,cm^{-1}$ at 100 °C. Long term stability test showed negligible performance degradation, and even at 30 °C, the proton conductivity remained at high level of $1.2 \times 10^{-2}\,S\,cm^{-1}$. Molecule dynamics (MD) simulations and X-ray total scattering experiments reveal the **HOF-g** system is consisted of three kinds of clusters, i.e., 1,5-Naphthalenedisulfonic acid (1,5-NSA) anion clusters, N,N-dimethylformamide (DMF) molecule clusters, and $H^+\text{-}H_2O$ clusters. In which, the $H^+$ plays an important role to bridge these clusters and the high conductivity is mainly related to the $H^+$ on $H_3O^+$. These findings provide valuable insights for optimizing HOFs, enabling efficient proton conduction, and advancing energy conversion and storage devices.

In recent years, hydrogen-bonded organic frameworks (HOFs) have attracted much attention as novel multifunctional materials due to their advantages such as easy synthesis, precise single-crystal structure, and low energy consumption in the regeneration process[1-7]. These frameworks offer potential applications in various fields, including catalysis[8-11], sensors[12-16] and gas adsorption[17-21] through rational structure design[5,22]. Specifically, HOFs are believed to hold great potential in proton conduction due to their rich inherent hydrogen bonding networks. However, a major challenge in harnessing the full potential of HOFs as efficient proton conductors lies in the presence of inter-particle boundaries, which hinders desired proton transport.

Conventionally, most studies have relied on the incorporation of guest molecules, such as water, to enhance the grain contacts and achieve high proton conductivity in HOFs[23-30], which is also common in other crystalline materials such as MOFs[31-37], COFs[38-45]. The guest molecules serve as mediators that facilitate proton transfer between the HOF particles. While this approach has shown promising results, it introduces practical limitations, particularly at elevated temperatures where managing humidity becomes cost-intensive and logistically challenging in fuel cell applications. With references to the elimination of grain boundaries in coordination polymers[46-55], vitrification is considered as one feasible method. Mason group[56] has done pioneering work and demonstrated that a desymmeterization strategy can be used to realize glass transitions of an alkylguanidinium sulfonate HOF. However, it is presumably that the entire HOF precursor is in a low energy state and the glassy material obtained does not exhibit significant proton conduction properties.

[1]Institute of Crystalline Materials, Shanxi University, Taiyuan 030006 Shanxi, China. [2]College of Chemistry, Taiyuan University of Technology, Taiyuan 030024, China. [3]School of Materials and Chemical Engineering, Zhengzhou University of Light Industry, Zhengzhou 450001, China. ✉e-mail: jtcl@sxu.edu.cn

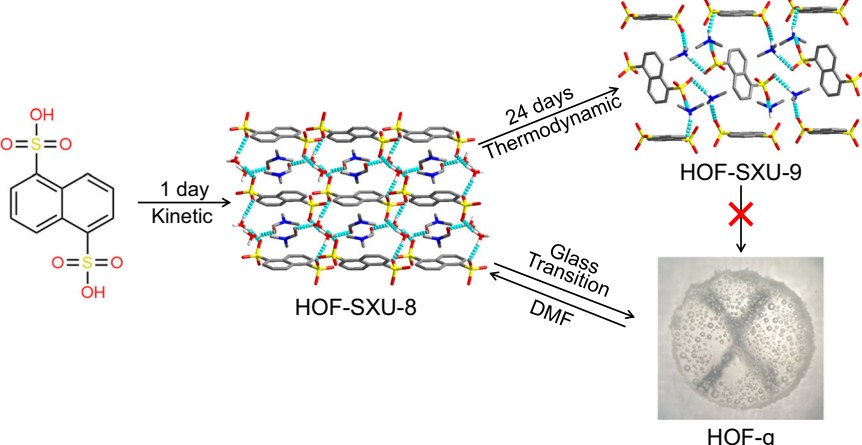

**Fig. 1 | Schematic representation of the synthesis of HOF-SXU-8, HOF-SXU-9 and the vitrification transformation process to HOF-g.** (Color code: C atom, gray; O atom, red; S atom, yellow; N atom, blue).

In this work, two single crystals, kinetically stable **HOF-SXU-8** with higher energy and thermodynamically stable **HOF-SXU-9** with lower energy were synthesized through liquid-phase diffusion method. **HOF-SXU-8** could undergo a vitrification transformation to glassy state **HOF-g** at around 110 °C (Fig. 1). Following this vitrification transformation, the proton conductivity of **HOF-g** without humidity at 100 °C reached as high as $5.62 \times 10^{-2}$ S cm$^{-1}$, while the conductivity for **HOF-SXU-8** is only $1.31 \times 10^{-7}$ S cm$^{-1}$. Remarkably, even at 30 °C, the proton conductivity of **HOF-g** maintained a quite high level of $1.2 \times 10^{-2}$ S cm$^{-1}$. Furthermore, after a stability test of 16 h or 5 rounds heating-cooling circles, the performance of **HOF-g** remained almost unchanged. On the other hand, due to its thermodynamic stability and limited free volume, **HOF-SXU-9**, does not undergo such transition.

## Results

### Crystal structure descriptions of HOF-SXU-8 and HOF-SXU-9

Single crystal X-ray analysis reveals that **HOF-SXU-8** crystallizes in the space group $P2_1/n$ (please see Supplementary Tab. 1 for more details), with 1,5-NSA anion layers separated by DMF molecules. $H_3O^+$ connects a continuous hydrogen bonding network capable of proton conduction by linking different 1,5-NSA anion layers and the DMF molecules (Supplementary Fig. 1). Specifically, O(A) on $H_3O^+$ forms one hydrogen bond with O(1 A) from the upper layer of 1,5-NSA anion (O·H···O, 2.65 Å, 166.85°) and another hydrogen bond with O(2 A) from the lower layer of 1,5-NSA anion (O·H···O, 2.65 Å, 152.37°), and one hydrogen bond with O(3 A) on DMF (O·H···O, 2.44 Å, 161.86°) as depicted in Supplementary Tab. 3. After soaking **HOF-SXU-8** in the mother solution for 4 days, the **HOF-SXU-8** crystals disappeared. Another 20 days later, thermodynamically stable colorless single crystals of **HOF-SXU-9** appeared in the space group $P2_1/c$ (please see Supplementary Tab. 2 for more details). Dimethylamine cations in **HOF-SXU-9** link the differently oriented 1,5-NSA anion through two hydrogen bonds (Supplementary Fig. 2 and Supplementary Tab. 4). The purity of bulk crystalline **HOF-SXU-8** and **HOF-SXU-9** were confirmed through good accordance with the powder X-ray diffraction (PXRD) pattern simulated from the crystallographic information files, as shown in Supplementary Fig. 3. Thermogravimetric (TG) analysis (Supplementary Fig. 4) revealed that the kinetically stable **HOF-SXU-8** starts to lose weight from 100 °C with relatively low thermal stability. In contrast, the thermodynamically stable **HOF-SXU-9** can keep its structure intact to about 310 °C. Above 310 °C, 1,5-NSA anion starts to decompose, leading to the collapse of its skeleton. Overall, thermodynamically stable **HOF-SXU-9** exhibits higher thermal stability.

### Crystal melting and glass formation

**HOF-SXU-8** was heated at 110 °C for 2 h and then quenched rapidly to obtain **HOF-g** in glassy form (details please see Synthesis of Methods section). To confirm the occurrence of vitrification, we used differential scanning calorimeter (DSC) technology to monitor the enthalpy response of the phase transition from -10 to 145 °C (Fig. 2a). When the first heating cycle was applied, a clear heat absorption peak appeared at 126 °C ($T_m$) and the appearance of which was attributed to the phase transition of solid-liquid transformation occurring in the crystal. The first cooling process in DSC confirmed the vitrification of **HOF-SXU-8** to a glassy state of **HOF-g** with a glass transition temperature ($T_g$) of 8 °C (Fig. 2b). The PXRD pattern of **HOF-g** exhibits no Bragg diffraction (Fig. 2c)[51–54], which is typical characteristic of glass phase. In the second cycle, the heat absorption peak disappeared, also confirming the occurrence of vitrification, and the glass transition temperature $T_g$ was calculated to be 9 °C (Supplementary Fig. 5). The minor differences in $T_g$ during the two rounds of cooling are attributed to the small changes of the microstructure in the second heating and cooling circle. DSC scans of crystalline **HOF-SXU-8** from -10 to 110 °C in Supplementary Fig. 6 demonstrate incomplete melting behavior. Although the accurate melting temperature $T_m$ of **HOF-SXU-8** is 126 °C, we chose to treat the **HOF-SXU-8** pellet at 110 °C for the glass transition because at this temperature **HOF-SXU-8** did not melt completely, allowing the shape of which to be better maintained. When heated to 200 °C, the guest molecules in the structure were totally removed, and the glass transition does not occur (Supplementary Fig. 7). PXRD analyses conducted on the **HOF-SXU-9** sample after heated at 110 °C for a duration of 2 h have revealed that the distinctive peaks were still remained (Supplementary Fig. 8), further suggesting that the transformation of **HOF-SXU-9** into an amorphous or glassy state did not occur.

Scanning electron microscope (SEM) image clearly shows the presence of grain boundaries in the compressed **HOF-SXU-8** pellet. The morphology of the HOF crystals varies, with well-defined edges and grain boundaries clearly seen in Fig. 2e. While for **HOF-g**, the grain boundaries disappear and the whole material exists in homogeneous form (Fig. 2f). Optical photography is an effective way to observe changes in samples during heat treatment, thus allowing observation of processes like melting[51,57]. **HOF-SXU-8** pellet exhibited different degrees of transparency after processed at different temperatures. In Fig. 2g, where the sample is placed on the upper side of a printed "X". The "X" is only visible after treated the pellet at 110 °C for 2 h and then cooled. Combining the results of the TG analyses, it is clear that the change in transparency is due to the occurrence of vitrification.

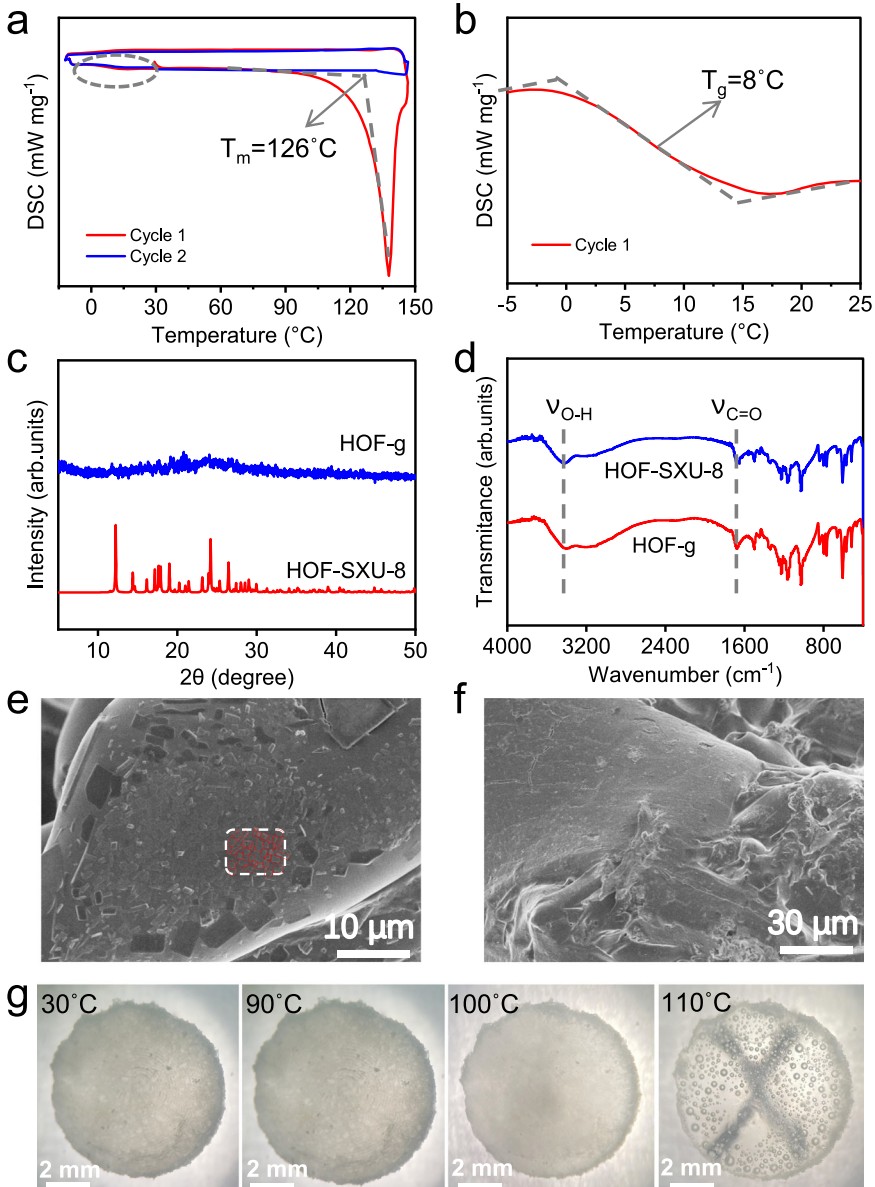

**Fig. 2 | Characterizations of the glass transition. a** DSC scans of crystalline **HOF-SXU-8** (Red curves represent the first heating and cooling cycle, blue curves represent the second cycle). **b** Enlarged Figure of the circled part in Fig. 2a. **c** PXRD patterns of **HOF-SXU-8** and **HOF-g**. **d** FTIR spectra of **HOF-SXU-8** and **HOF-g**. **e** SEM image of the **HOF-SXU-8** pellet. **f** SEM image of the **HOF-g**. **g** Optical photographs of crystalline **HOF-SXU-8** after treated at different temperatures. The sample was put on the mark "X" to assess the variation in transparency.

There are many bubbles in the glass pellet, which we speculated originated from the vaporization of low boiling point products from the breakdown of DMF in **HOF-SXU-8**, based on the fact that DMF is easy to decompose under conditions of strong acidity (-SO$_3$H in **HOF-SXU-8** provide) and elevated temperatures[58]. **HOF-g** was observed to re-crystallize (Supplementary Fig. 9) back to **HOF-SXU-8** under heat treatment in DMF atmosphere, which is proved by the PXRD results (Supplementary Fig. 10). However, no such re-crystallization was observed in the absence of DMF atmosphere, regardless of the same heat treatment temperature. In Fig. 2d, the C = O (1680 cm$^{-1}$) stretching vibration comes from the DMF structural unit and the O-H (3413 cm$^{-1}$) stretching vibration comes from the H$_3$O$^+$ structural unit, which is bound to the O-H bond after the formation of hydrogen bonding and the spectral peak broadens[59]. These characteristic peaks are present in the Fourier transformed infrared (FTIR) spectra of both **HOF-SXU-8** and **HOF-g**, proving that the DMF structural unit and the hydrogen

bonding network are still present in the structure after the vitrification process (Fig. 2d). Liquid $^1$H NMR spectroscopy results illustrate that **HOF-g** loses about 30% of DMF compared to **HOF-SXU-8** (Supplementary Figs. 11 and 12). TG result reveals that in the range of 100–150 °C, the weight loss of **HOF-SXU-8** is about 7.1% (Supplementary Fig. 4a), corresponding to 30% DMF of **HOF-SXU-8**. We have further designed two rounds of heating-cooling cycles on **HOF-SXU-8** to monitor its weight loss behavior (Supplementary Fig. 13). First, **HOF-SXU-8** was heated to 110 °C, kept at 110 °C for 2 h, and then cooled to room temperature. This process resulted in about 7.3% weight loss of the sample, corresponding to the loss of 30% DMF in **HOF-SXU-8**. During the second heating-cooling round, the sample was heated from room temperature to 160 °C and then cooled to room temperature. During this round, there was no weight loss observed in the temperature range of 100–150 °C. Weight loss continued only at temperatures above 153 °C, which surpasses the boiling point of DMF.

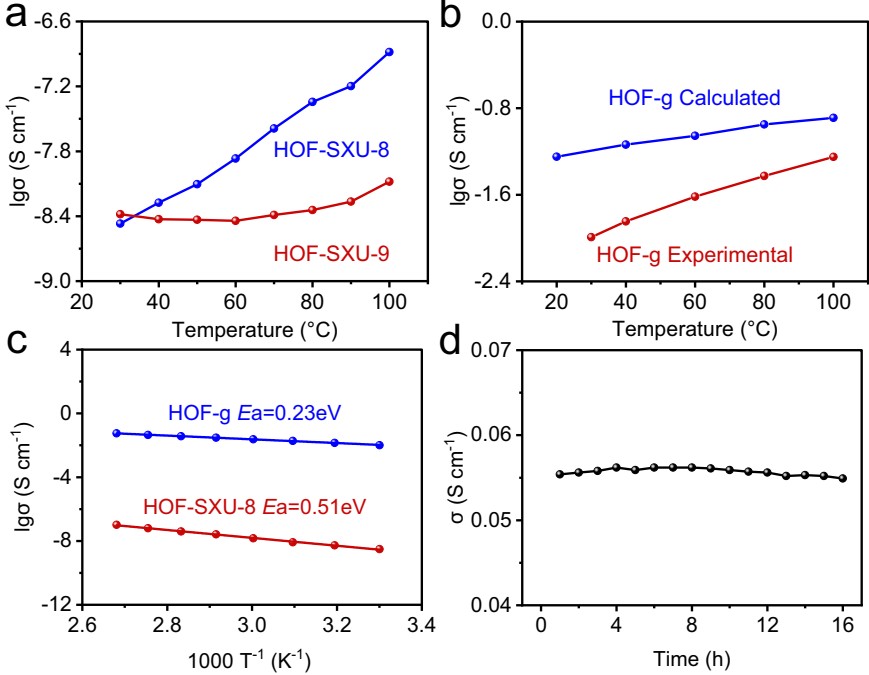

**Fig. 3 | Proton conductivity property of HOF-SXU-8, HOF-SXU-9 and HOF-g.**
**a** Proton conductivity of **HOF-SXU-8** and **HOF-SXU-9** with increasing temperature.
**b** Experimental (red) and calculated (black) proton conductivity of **HOF-g** with
increasing temperature. **c** Arrhenius plot of **HOF-SXU-8** and **HOF-g**. **d** Time-dependent proton conductivity of **HOF-g** performed at 100 °C.

This indicates that after keeping at 110 °C for 2 h, the decomposition of DMF reached a maximum. Elemental analysis (EA) showed that the molecular formulas were $(1,5\text{-NSA})_{1.04}$ $(H_3O)_{2.25}$ $(DMF)_{2.2}$ for **HOF-SXU-8** and $(1,5\text{-NSA})$ $(H_3O)_{2.26}$ $(DMF)_{1.56}$ for **HOF-g** (Supplementary Tab. 5), with the difference related to the loss of ~30% DMF after glass formation. The above results demonstrate that the glass transition occurs due to the decomposition of 30% DMF in the structure, which is different from the complete guest molecules removal as previously reported by Mason's group[56].

To verify the state of **HOF**-g, the mechanical properties of **HOF-g** were evaluated from viscosity and dynamic mechanical analyses (DMA). The viscosity values (Supplementary Fig. 14) of **HOF-g** at 30-120 °C are lower than the viscosity at the Littleton softening point ($10^{6.6}$ Pa s), indicating that **HOF-g** exhibits some processing ability. Further increasing the temperature, the fluidity of **HOF-g** will rapidly become larger. The DMA measurement performed in Supplementary Fig. 15 is an upscan process. In Supplementary Fig. 15, the storage modulus (G') was observed to dominate over the loss modulus (G") from 30 to 120 °C, indicating the lack of fluidity in HOF glass even above the glass transition temperature. The immediate reduction of G' above 120 °C signifies the softening of **HOF-g**, as previously discussed in reported publications[47,49]. As for the lack of observable changes in G' and G" near the glass transition point, this can be attributed to the relatively stable molecular structure of the material below and above the glass transition temperature. The material does not undergo significant structural changes in response to temperature variations, resulting in minimal alterations in both G' and G". Furthermore, the G' and G" of the sample exhibit low sensitivity to temperature changes, which contributes to the observed stability near the glass transition point. Our analysis of the G' and G" curves around 150 °C suggested that HOF glass resides between a high elastic state and a viscous flow state near this critical temperature point. Upon increasing the temperature to 160 °C, G' drastically decreased from ~$10^6$ Pa to about $10^2$ Pa due to structural damage to the HOF glass[60,61].

## Proton conduction properties

Electrochemical impedance spectroscopy (EIS) of **HOF-SXU-8, HOF-SXU-9** and **HOF-g** were measured without humidity at different temperatures. The conductivity values of **HOF-SXU-8** exhibit an increase from $3.40 \times 10^{-9}$ S cm$^{-1}$ at 30 °C to $1.31 \times 10^{-7}$ S cm$^{-1}$ at 100 °C (Fig. 3a and Supplementary Fig. 16) with the crystalline phase of **HOF-SXU-8** remained throughout the measurements (Supplementary Fig. 17). Crystalline structure of **HOF-SXU-9** was also maintained (Supplementary Fig. 17) and the conductivity slightly increased from $4.15 \times 10^{-9}$ S cm$^{-1}$ at 30 °C to $8.31 \times 10^{-9}$ S cm$^{-1}$ at 100 °C (Supplementary Fig. 18), which is ranked at a very low level[28,62]. This phenomenon may be related to the strong hydrogen bonding network in the thermodynamically stable **HOF-SXU-9**.

Conductivity measurements of **HOF-g** revealed a high value of $1.02 \times 10^{-2}$ S cm$^{-1}$ at 30 °C, and then increased slowly to $5.62 \times 10^{-2}$ S cm$^{-1}$ at 100 °C (Fig. 3b and Supplementary Fig. 19). The proton conductivity of **HOF-g** has been improved by about 5 orders of magnitude compared to single-crystalline **HOF-SXU-8**, making **HOF-g** a superior proton conductive material (Supplementary Tab. 6)[24,26,27,29,63–70]. The Arrhenius plot (Fig. 3c) demonstrates different mechanisms of **HOF-SXU-8** and **HOF-g**: the activation energy of **HOF-SXU-8** was calculated to be 0.51 eV for the vehicular mechanism, while the activation energy value of **HOF-g** was 0.23 eV, which is typical for the Grotthuss mechanism[71]. In addition, **HOF-g** can maintain its superior proton conductivity up to 16 h at 100 °C (Fig. 3d and Supplementary Fig. 20) with performance kept almost the same, revealing that the glassy **HOF-g** structure is well retained in this process. Even after five rounds of proton conductivity tests, the proton conductivity can still be maintained at ~$5 \times 10^{-2}$ S cm$^{-1}$ at 100 °C (Supplementary Fig. 21). This suggests that the glassy **HOF-g** sample is under equilibrium and exhibits robust stability at 100 °C. However, beyond 150 °C, an obvious decline in proton conductivity is observed (Supplementary Fig. 22) due to the simultaneous departure of guest molecules confined in **HOF-g** including DMF and $H_2O$, which is consistent with the TG results

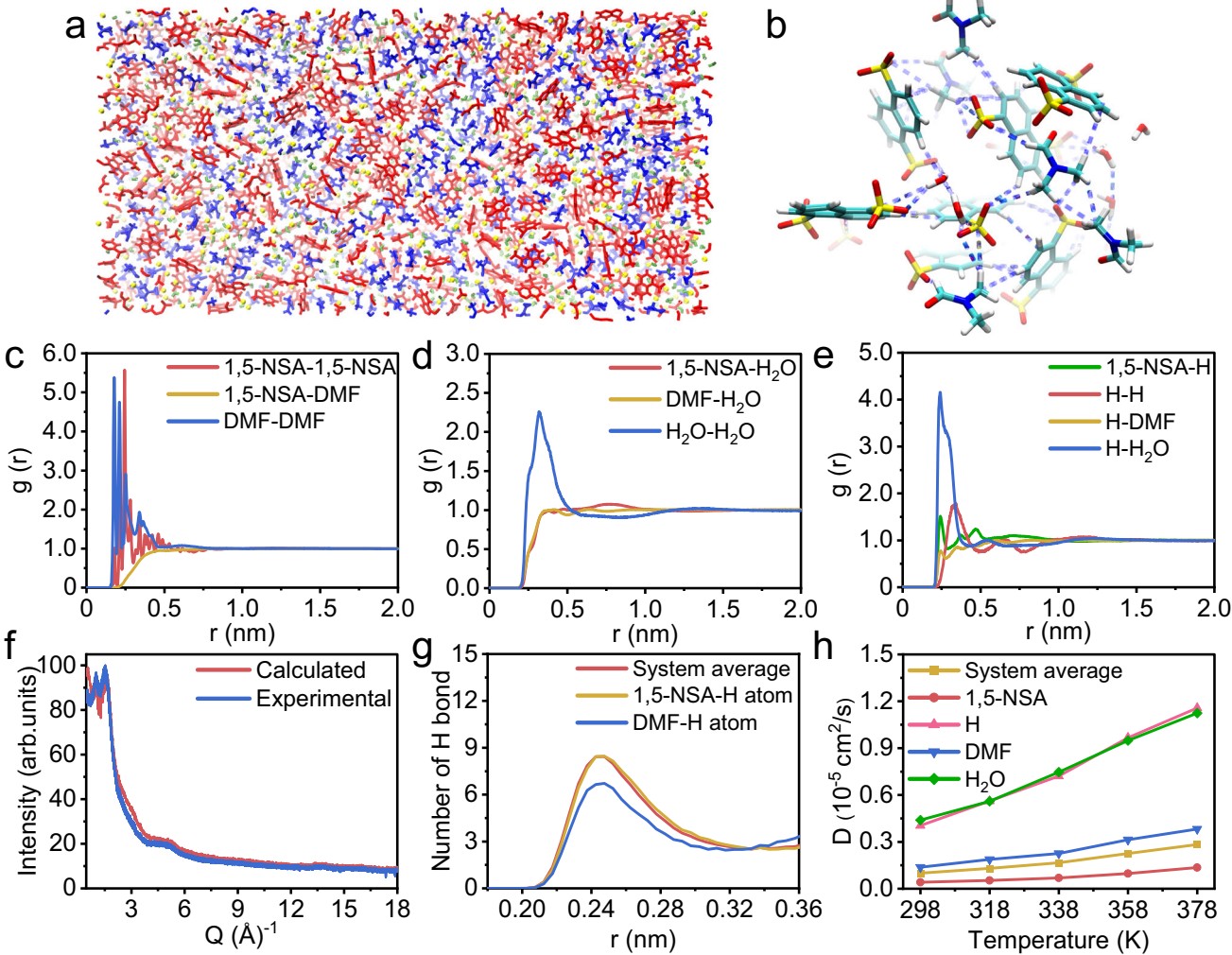

**Fig. 4 | The proton conduction mechanism of HOF-g. a** Structure diagram of simulated **HOF-g** (Cluster Color: 1,5-NSA anion, red; DMF molecule, blue; H⁺, yellow; H₂O molecule, green). **b** Transient local structure of simulated **HOF-g** in MD simulation process (Color code: C atom, pale blue; O atom, red; S atom, yellow; N atom, blue; H atom, white). **c** RDFs associated with 1,5-NSA anion and DMF molecules. **d** RDFs associated with H₂O molecules. **e** RDFs associated with H ions. **f** X-ray total scattering pattern of **HOF-g** at room temperature (blue) compared with that calculated from MD simulations (red). **g** Distribution of the number of hydrogen bonds over distance. **h** Self-diffusion coefficients of different components from **HOF-g** at different temperatures.

(Supplementary Fig. 13). This phenomenon indirectly indicates that the carrier of proton conductivity is related to H⁺ in H₃O⁺.

## Simulation and proton conduction mechanism

In order to probe the proton conduction mechanism, we have performed MD simulations using LAMMPS 2022 software (details please refer to methods section). An energy minimization procedure and an NVT ensemble simulation ($T = 298$ K, $p = 1$ bar, time = 1 ns) were employed to obtain equilibrated conformation (Fig. 4a). Further NVT ensemble simulation (15 ns, T = 298 K) and annealing procedure (5 ns, 298 K-873 K) were performed to assure that the above obtained system is stable.

The radial distribution functions (RDFs) of 1,5-NSA anion-1,5-NSA anion, DMF-DMF have similar multiple peaks within 0.6 nm (Fig. 4c), suggesting that both 1,5-NSA anion and DMF spontaneous aggregated to clusters with a radius of 0.5-0.6 nm and exhibited a certain spatial distribution. The RDF of 1,5-NSA anion-DMF is low within 0.6 nm, indicating that the two clusters are mutually exclusive. Beyond 0.8 nm, it gradually converges to 1, indicating that the clusters are more uniformly dispersed with each other. Besides, there are significant interactions between the H₂O molecules to form clusters with radius of 0.4–0.5 nm, while no significant interactions between H₂O and 1,5-NSA

anion, or between H₂O and DMF (Fig. 4d). Beyond 1.2 nm, the RDFs associated with H₂O converge to 1, indicating that the H₂O clusters are dispersed with other components of **HOF-g** (Fig. 4d). Similarly, the RDF between H⁺ and H₂O (Fig. 4e) is exceeded near 0.36 nm, suggesting that an indirect association is formed between H⁺ relying on the H₂O molecule, while no significant H⁺-1,5-NSA anion and H⁺-DMF interaction were observed. In summary, the system is summarized by the following features: (1) the substances are divided into three kinds of clusters, i.e., 1,5-NSA anion clusters, DMF molecule clusters, and H⁺-H₂O clusters, with the cluster radius of 0.4–0.6 nm; (2) the clusters are diffused with each other over medium and long distances (>0.8–1.2 nm). We also obtained the X-ray total scattering pattern of **HOF-g** and compared it with the simulated data based on the MD model (Fig. 4f). The experimental X-ray total scattering pattern of **HOF-g** agrees well with that of the simulated results, indicating that the structure of **HOF-g** is consistent with simulated **HOF-g**, further confirming the accuracy of our simulated **HOF-g** structure as discussed above[72–74]. The X-ray total scattering data obtained from experimental data was also corrected, normalized, and Fourier transformed to obtain a reduced radial distribution function (Supplementary Fig. 24). Below 8 Å, the experimental PDF pattern shows sharp peaks, suggesting that the **HOF-g** glass might be ordered at short distances.

Similar sharp peak features are also observed in the PDFs of ZIF glasses[75]. While beyond 8 Å, the experimental PDF exhibits much weaker peak, indicating the disorder in long-range.

The final equilibrated structure was then subjected to 1 ns equilibrium sampling, with the structure saved every 5 ps to form a trajectory and the interaction energy saved every 1 ps. The electrostatic interaction energies and long-range non-bonding interaction energies between 1,5-NSA anion, DMF molecules, $H^+$, and $H_2O$ were decomposed, and Supplementary Fig. 25 demonstrates the interaction energies of each group as a percentage of the total interaction energy. 1,5-NSA anion-1,5-NSA anion, 1,5-NSA anion-$H^+$, DMF-DMF, $H^+$-$H^+$, and $H^+$-$H_2O$ interactions occupied the top four proportion. 1,5-NSA anion-$H_2O$ had a lower interaction energy (4%), whereas 1,5-NSA anion-$H^+$ (19.7%) and $H^+$-$H_2O$ (8.8%) had higher interaction energies, so $H^+$ acted as a bridge connecting 1,5-NSA anion clusters to $H_2O$ clusters, leading to the two types of clusters mixed and dispersed. Identically, 1,5-NSA anion-DMF had a lower interaction energy (0.9%), while the interaction energies of 1,5-NSA anion-$H^+$ and $H^+$-DMF account for 19.7% and 3.8%, respectively, thus $H^+$ also serves as a bridge between 1,5-NSA clusters and DMF clusters. In short, among the three original constituents of the **HOF-g** system, $H^+$ acts as a bridge between the clusters of the whole system.

The snapshot of the local structure in the middle period of simulation was drawn in Fig. 4b to display the rich hydrogen bonding. Hydrogen bonding analysis was carried out on the simulated trajectories to elucidate the contribution of different components. The number of hydrogen bonds generated by 1,5-NSA anion, DMF molecules, and the system averaged over different distances was analyzed (Fig. 4g). The hydrogen bonds formed by 1,5-NSA anion and DMF molecules in the system were distributed mainly at the distance range of 0.22–0.28 nm, and the number of hydrogen bonds formed by 1,5-NSA anion was significantly higher than that of DMF molecules. Thus, for this system, more hydrogen bonds can be formed in the same volume due to the release of DMF, which strengthens the hydrogen bonding network in the system and forms a glassy structure, and in turn would greatly enhance the proton conductivity of the glassy **HOF-g**.

The self-diffusion coefficients of different components in the simulated **HOF-g** increased with the increasing of temperature (Fig. 4h). The self-diffusion coefficients of $H^+$ and $H_2O$ are the largest two among all species and are always closer at different temperatures, which also confirms the existence of $H^+$-$H_2O$ clusters. The calculated conductivity values of the simulated **HOF-g** system at different temperatures are shown in Fig. 3b. At 308 K and without an applied electric field, the conductivity is $6.41 \times 10^{-2}\,S\,cm^{-1}$, which is quite close to the experimentally obtained conductivity of $1.20 \times 10^{-2}\,S\,cm^{-1}$. The deviation of the proton conductivity between experimental and calculated proton conductivity falls in the same order of magnitude, which is acceptable[76,77]. Due to the huge difference in the self-diffusion coefficients between $H^+$ and 1,5-NSA anion, the main contribution to the electrical conductivity (90.7%) of the system comes from the diffusion of hydrogen ions ($H^+$-$H_2O$ clusters) (Fig. 4h).

## Discussion

In summary, we have successfully prepared two hydrogen bonded organic frameworks: kinetically stable **HOF-SXU-8** and thermodynamically stable **HOF-SXU-9**, through controlling the reaction time. **HOF-SXU-8** could be transformed at about 110 °C to glassy state **HOF-g** triggered by partial decomposition of DMF molecules. **HOF-g** exhibits superior proton conduction performance from 30 °C to 100 °C without humidity, with conductivity reaching as high as $5.62 \times 10^{-2}\,S\,cm^{-1}$ and excellent stability. MD simulations and X-ray total scattering experiments reveal the **HOF-g** system can be divided into three kinds of clusters, i.e., 1,5-NSA anion, DMF molecule, and $H^+$-$H_2O$ clusters. In which, the $H^+$ plays important role to bridge these clusters and the conductivity is mainly related to $H^+$ on $H_3O^+$. We believe that the glassy HOF will be a design well worth exploring in depth and may open new avenues for obtaining safe and high-performance pure solid-state electrolytes.

## Methods

### Synthesis

All solvents and materials were purchased commercially and used without further purification. 1,5-NSA (Specification and purity:99%; contain water ≤ 15%) was purchased from Adamas-Beta. DMF and *n*-hexane (AR, ≥99.5%) were purchased from Tianjin Damao Chemical Reagent Factory.

Synthesis of **HOF-SXU-8**: 1,5-NSA (2.08 mmol, 600 mg) was dissolved in DMF (5 mL) in a 20 mL glass vial by ultrasound at 25°C. 1 mL of the above solution was then placed in a 20 mL glass vial and 2 mL hexane was added slowly. After 1 day, transparent **HOF-SXU-8** crystals were precipitated in the bottom of the glass vial. Crystallographic data for **HOF-SXU-8** has been deposited with the Cambridge Crystallographic Data Centre, CCDC, depository number 2292290. [1]H NMR (600 MHz, DMSO-$d_6$) δ 8.86 (d, 2H), 7.94 (m, 3H), 7.43 (dd, 2H), 2.89 (s, 3H), 2.73 (s, 3H).

Synthesis of **HOF-SXU-9**: The above obtained **HOF-SXU-8** was observed to disappear after 4 days of its appearance. Another 20 days later, thermodynamically stable crystals of **HOF-SXU-9** appeared in the bottom of the glass vial. Crystallographic data for the **HOF-SXU-9** has been deposited with the Cambridge Crystallographic Data Centre, CCDC, depository number 2292291.

Synthesis of **HOF-g**: **HOF-g** were obtained by rapid melt quenching of **HOF-SXU-8**. **HOF-SXU-8** crystals were ground to powder first and 15 mg of the powder was pressed into a pellet. The pressed **HOF-SXU-8** pellet was further heated in an oven at 110 °C for 2 h and then quickly quenched to obtain **HOF-g**. [1]H NMR (600 MHz, DMSO-$d_6$) δ 8.86 (d, 2H), 7.95 (m, 3H), 7.43 (dd, 2H), 2.89 (s, 3H), 2.73 (s, 3H).

### Characterizations

TG was performed on a NETZSCH STA 449 C thermal analyzer under nitrogen ($N_2$) atmosphere from 30 to 900 °C at a rate of 10 °C/min. $T_g$ was determined by differential scanning calorimetry (DSC) (STA 404 C,Netzsch) and collected with a Netzsch DSC 200 F3 under $N_2$ atmosphere (Al crucible). The morphology of **HOF-SXU-8** and **HOF-g** were analyzed using SEM (Zeiss EVO 60 S). The changes of the samples after heat treatment were observed by optical and microscope. [1]H MAS Solid-state NMR was measured by the single pulse technique with a relaxation time of 3.00 s at a 10 kHz spinning rate ([1]H resonance frequency of 599.72 MHz). Elemental analyses (C, H, S and N) were performed in-house using an Elementar vario EL elemental analyzer. Dynamic mechanical analysis and viscosity measurement were evaluated via a rotational parallel-plate rheometer (DHR-2), applying oscillatory strain of 1 Hz.

### X-ray characterization

The single crystal data were collected with a Rigaku XtaLab P200 diffractometer and a Dectris Pilatus 200 K system at 302 K. The system was equipped with A MicroMax007 HF/VariMax rotating anode X-ray generator with confocal monochromatic Mo-Kα radiation. The collected data were solved and refined by full matrix least squares using SHELXL 2016/4. The phases of the synthesized crystals were identified using powder X-ray diffraction (PXRD) method. The PXRD experiments were performed on a Rigaku Ultima IV X-ray diffractometer at 40 kV and 40 mA with a scan rate of 1°/min and a 2θ range of 5–50°, and the diffractometer was equipped with a Cu sealed tube (λ = 1.5406 Å). Based on the single crystal diffraction data, simulated PXRD patterns were generated using Mercury 3.9 software.

## Proton conductivity measurements

Proton conductivity was characterized through a two-probe method using a Solartron SI1260 Impedance / Gain-phase analyzer coupled with a Solartron 1287 dielectric interface. Before conductivity measurements, **HOF-SXU-8** and **HOF-SXU-9** samples were placed in a steel mold and pressed into pellets of 6 mm diameter and 1.5 mm thickness using a tablet press. For the measurement of **HOF-g**, 100 mg **HOF-SXU-8** was first pressed into a 6 mm diameter pellet (pressure 10 MPa). **HOF-SXU-8** pellet was then put into a 2032 steel cell in a glove box filled with ultra pure Ar ($H_2O$ and $O_2 < 1$ ppm) and heated in an oven at 110°C for 2 h to obtain **HOF-g** by rapidly melt quenching. All measurements in this work were performed under a stream of ultra-pure $N_2$ (>99.99%). The measurements were conducted in the frequency range from 0.5 Hz to 1 MHz with an input voltage amplitude value of 100 mV. The impedance values at each temperature were repeatedly measured after 30 min of equilibrium until the measured values keep stable. The resistance values were obtained by fitting the impedance profile using Zview software. The circuit equivalent used for fitting is as follows:

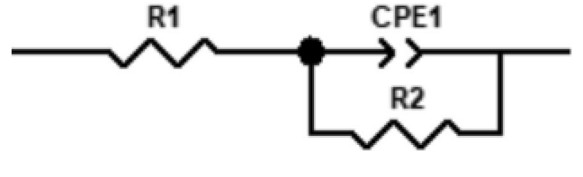

$$\sigma = 1/RS \tag{1}$$

$$\sigma T = A \exp\left(-E_a/k_B T\right) \tag{2}$$

R1 corresponds to the resistances of wire and electrode, while R2 accounts for the bulk resistance of the pellet. CPE1 denotes the nonideal capacitance corresponding to the bulk. The conductivity was calculated using Eq. (1), where $\sigma$ is the conductivity (S $cm^{-1}$), $l$ is the pellet thickness (cm), $S$ is the solid electrolyte area ($cm^2$), and $R$ is the bulk resistance of the pellet (R2 in the circuit equivalent) fitted by the equivalent circuit of the semicircle in Nyquist plot using zview software. The activation energy ($E_a$) of the material conductivity is estimated from Eq. (2), where $A$ is the exponential prefactor, $k_B$ is the Boltzmann constant, and $T$ is experiment temperature.

## X-ray total scattering measurements

**HOF-SXU-8** sample was packed in a borosilicate capillary with a diameter of 0.7 mm, then in-situ heated at 110°C and quenched rapidly back to room temperature to obtain **HOF-g** in the borosilicate capillary. The X-ray total scattering data was collected on the Mython-II detector over 115° covering the Q range up to 18 Å$^{-1}$ (21 keV; $\lambda = 0.5921$ Å). The wavelength, zero error and instrument contribution to the peak profile was determined using the line position and line shape standard NIST Si sample with the refined wavelength 0.5921 Å. For the experiment, an empty capillary was used as the background. The collected scattering data was applied absorption, background, and Compton scattering corrections then normalized to give the X-ray total scattering pattern.

## Computational method

Molecular dynamics simulations were employed to study the material. The simulation is based on GAFF force field, and the force field parameters are obtained from quantum chemistry calculation and fitting, using m2seminario method. The long-range electrostatic term in the force field is calculated using Particle-Mesh-Ewald (PME) method, and the van der Waals interaction is calculated using Lennard-Jones potential (cutoff = 1.0 nm).

The simulation system contains 1440 naphthalenedisulfonic molecules, 2016 DMF molecules and 2880 water molecules, 62464 atoms in total.

Firstly, energy minimization procedure and an NVT ensemble simulation ($T = 298$ K, $p = 1$ bar, time = 1 ns) were employed to obtain a equilibrated and compressed conformation. After these processes, the system has a volume of $7.10 \times 10^5$ Å$^3$ and density of $1.26 \times 10^3$ kg/m$^3$.

Secondly, an NVT ensemble simulation (15 ns, $T = 298$ K) and an annealing procedure (5 ns, 298 K-873 K) were performed to assure that the system has been stable. The annealing procedure were performed in the protocol as below: 100 ps for each cycle, heat from 298 K to 873 K for 30 ps, keep temperature in 873 K for 30 ps, anneal from 873 K to 298 K in 5 ps, and keep temperature in 298 K for 35 ps, repeat the cycle for 50 times. After the above annealing procedure, the end structure kept the same as the start structure, which indicates that the system has been the stable structure already (Supplementary Fig. 23).

A sampling for 1 ns was performed to the productive simulation, in which the timestep is 1 fs, trajectories were recorded every 5 ps, energies and properties were recorded every 1 ps.

## Data availability

The data generated in this study are provided in the Source Data file. Source data are provided with this paper. The X-ray crystallographic coordinates for structure reported in this article have been deposited at the Cambridge Crystallographic Data Centre (CCDC), under deposition number CCDC 2292290 and 2292291. These data can be obtained free of charge from The Cambridge Crystallographic Data Centre via www.ccdc.cam.ac.uk/data_request/cif. Additional data are available from the corresponding author upon request. Source data are provided with this paper.

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

## Acknowledgements

This research was supported by National Natural Science Foundation of China (22001154 L.L.).

## Author contributions

F.Y. performed most of the experiments and analyzed data. X.W. performed proton conductivity measurements and analyzed the datas. J.T. performed a collection of single-crystal structures. Y.Y. performed proton conductivity measurements. L.L. conceived and designed the experiments and modified the paper. All authors discussed the results and commented on the manuscript.

## Competing interests

The authors declare no competing interests.
