## [Peer Review File · Nature Communications]

REVIEWER COMMENTS

Reviewer #1 (Remarks to the Author):

The authors used HOF and its crystalline melting and vitrification by cooling. They report that the proton-conducting properties of the glass are significantly higher than those of the crystalline phase. The paper has some major problems in the scientific discussion.

1. The melt quenched glass shown in Figure 1g has a significant amount of bubbles in the pellet. This suggests irreversible decomposition or transformation at higher temperatures. DSC profiles up to about 140-150 degC and holding this temperature for a few hours (at least) should show the clear peak of melting behaviour and no thermal degradation. It is also strange that no T_g is observed in the downscan of the DSC. The results of the elemental analysis alone are not sufficient to discuss these points.

2. The diameter of a 2032 steel coin cell is 20 mm. On the other hand, the samples used in the AC impedance measurements are 6 mm pellets which are not matched in size. When a sample is placed in a button cell, it is not possible to observe if the shape of the sample changes. How can the authors accurately determine sample area/height and subsequent conductivity after melting? The size and volume of such a soft compound changes greatly with temperature.

3. Related to point 1 above, the results in Figure 2d alone are unsatisfactory considering the stability and dullability of the sample. Cycle impedance measurements from low to high temperatures should be conducted several times. If they do not observe the same data set, it suggests that the measurements are not under equilibrium or that the sample shape is changing.

4. The mechanism of anhydrous proton conduction is still unclear: does the conductivity decrease when measured at temperatures above 100 degC? It is recommended to check the impedance at around 120 to 140 degC, which suggests (albeit indirectly) that the carriers of the conductivity are H⁺ and H₃O⁺.

5. Solid-state ¹H NMR has not been properly discussed. The authors should make a peak assignment of the HOF structure in the spectrum of Figure 3g before discussing the mobile proton peaks. It is not possible to describe the proton motion mainly from the chemical shifts. Peak sharpening should be observed at different temperatures or relaxation times should be measured.

Although they combine various characterisation and computational studies, I found significant scientific misleading in terms of glass conductivity and their interpretation, and do not recommend it for publication.

Reviewer #2 (Remarks to the Author):

Comments to the Authors

Yang and co-authors reported in this manuscript vitrification as an effective strategy for the kinetically stable HOF material HOF-SXU-8 to eliminate HOF grain boundaries and obtain significantly enhanced proton conductivity without humidity. They have also investigated the transformation process and the structure after the glass transition through various experiments combined with molecular dynamics simulations. It is pointed out that protons play a connecting role between different clusters of the glassy structure and have a higher mobility than in the original HOF and thus much higher proton conductivity. This paper is well organized and contributes a meaningful scheme for the elimination of HOF grain boundary effect, which enables significant enhancement of proton conductivity. The proposed scheme has a number of advantages over state of the art and is expected to further guide the development of more super-protonic conductors. However, there are some problems which must be solved before it is considered for publication. If the following problems are well-addressed, I believe that the contribution of this paper is important for HOF applications. Therefore, I recommend publication after minor revisions.

- 1, Several important reviews about design rules and functionalities of stable HOFs are missing in the introduction.
2. The authors propose that HOF-SXU-8 is a kinetic product and HOF-SXU-9 is a thermodynamic product, can they give a mechanistic explanation as to why this transformation has occurred. Also, why can't HOF-SXU-9, whose main structure consists of the same 1,5-NSA molecules, undergo similar glass transition like HOF-SXU-8?
- 3, In Figure 1g, there are many visible bubbles after processed at 110°C, please explain this. Do these bubbles have any effect on the conductive properties of the proton? I am wondering whether these bubbles can be removed to form a better homogeneous phase of the glass or not.
- 4, The authors claim that their HOF-based glass exhibit superior proton conductivity. It is necessary to list the proton conductivities of HOF-based materials in a special table for comparison.
- 5, In the reported research work on proton conductivity, cycling experiments on proton conductivity are usually included. This is important to measure whether a material can be practically applied. So does HOF-SXU-8-g maintain its performance after several rounds of warming and cooling experiments? Therefore, the authors should conduct additional experiments to verify this.

6, Some of the sentences have grammatical mistakes and need to be adjusted accordingly, such as, in Page 9, The sentence "the system are summarized" contains a subject-verb agreement error. It should be "the system is summarized" to ensure agreement between the singular subject ("the system") and the singular verb ("is"). The manuscript needs careful polishing and particular attention to English grammar, spelling, and sentence structure.

Reviewer #3 (Remarks to the Author):

The authors here reported two kinds of HOFs constructed by 1,5-NSA, DMF and H₂O. Specifically, HOF-SXU-8 can be processed glassy state HOF-SXU-8-g by rapidly quenching approach. In contrast to HOF-SXU-8 (1.31×10^{-7} S cm⁻¹), HOF-SXU-8-g represented significantly enhanced anhydrous proton conductivity (5.62×10^{-2} S cm⁻¹), long-term stability, and temperature insensitivity. MAS NMR and MD simulation reveals that conductivity is mainly related to H⁺ on H₃O⁺. Generally, the result is novel and interesting, and I enjoy the reading of this manuscript. Therefore, the publication of this manuscript in Nat. Comm is recommended, but the following questions need to be addressed:

1. The authors claimed that T_g is 20.66°C (It might be more appropriate to describe it as 21°C). Does it mean HOF-SXU-8-g is a softer liquid-like state rather than glass state at 30 °C that most characterizations were conducted? The evidence in Figure 1b is doubtful. Please provide more evidences to affirm the state of HOF-SXU-8-g.
2. Page 5 Line 113, the authors claimed that T_m is 100.26°C (It might be more appropriate to describe it as 100°C) and HOF-SXU-8-g has fewer DMF molecules in structure than HOF-SXU-8. The boiling point of DMF is 153°C and there are hydrogen bonding between DMF and H₂O in structure. Furthermore, based on the results from MD calculation, there are almost no interactions between water and DMF/1,5-NSA anion molecules in HOF-SXU-8-g. Why DMF can be removed at 110°C rather than H₂O? Apart from the evidence from elemental analysis, additional experiments, such as ¹H NMR, should be added to support authors' conclusion.
3. Since this material is not crystalline, how the authors can determine the molecular formulas? Please elucidate the detail procedure on establishing the molecular formulas of HOF-SXU-8-g . Furthermore, HOF-SXU-8-g has obvious difference between calculated and experimental elemental analysis for C. Please provide an explanation.
4. Please provide the table for comparing the proton conductivity between HOF-SXU-8-g and other reported materials.
5. Variable-temperature solid-state NMR should be added to provide insight into structural dynamics.
6. Add the fitting of experimental data from pair distribution function(PDF).
7. I can find an obvious difference between experimental and calculated proton conductivity. Please provide an explanation.
8. Page 6 Figure 1a, clarify why there are four lines for two samples?

9. Compared to T_g of ZIF glass (>400°C), why does HOF have a much lower T_g?

10. Page 8 Line 153, detailed simulation description is in “methods section” in Manuscript, not in SI.

Response to Reviewer 1 Comments

Reviewer 1:

Comments to the Author

The authors used HOF and its crystalline melting and vitrification by cooling. They report that the proton-conducting properties of the glass are significantly higher than those of the crystalline phase. The paper has some major problems in the scientific discussion.

Response: We appreciate your attention to our work and we extend our sincere thanks for your valuable feedback. According to the suggestions, we have thoroughly reviewed the results and discussion section and addressed the major concerns you have raised point by point. Your constructive suggestion is highly valued, and we have made the necessary revisions to ensure the quality of our manuscript. Thank you again for your time and insights. Corresponding revisions and point-to-point responses are listed as following:

Comment 1: The melt quenched glass shown in Figure 1g has a significant amount of bubbles in the pellet. This suggests irreversible decomposition or transformation at higher temperatures. DSC profiles up to about 140-150 degC and holding this temperature for a few hours (at least) should show the clear peak of melting behaviour and no thermal degradation. It is also strange that no Tg is observed in the downscan of the DSC. The results of the elemental analysis alone are not sufficient to discuss these points.

Response: Many thanks for your comments and suggestions. As demonstrated in the main text, the glass transformation occurs accompanying the loss of DMF, which resulted in enhanced local motion of the rest molecules. The large amount of bubbles was caused by the vaporization of low boiling point products from the breakdown of DMF in **HOF-SXU-8**, based on the fact that DMF is easy to decompose under conditions of strong acidity (-SO₃H in **HOF-SXU-8** provide) and elevated temperatures (Tetrahedron **2009**, 65(40), 8313-8323). Indeed, such decomposition is an irreversible process. We have also demonstrated experimentally that holding the system at either 110°C for 2 hours or at 145°C for 2 hours will similarly cause the **HOF-SXU-8** system lose roughly 30% DMF while undergoing similar glass transformation. Further weight loss does not occur until the temperature was further

increased above 153°C, which is higher than the boiling point of DMF. Specific details of these experiments are described below:

We have performed two rounds of heating-cooling cycles on **HOF-SXU-8** to monitor its weight loss behavior (Fig S13). First, **HOF-SXU-8** was heated to 110°C, kept at 110°C for 2 hours, and then cooled to room temperature. This process resulted in about 7.3% weight loss of the sample, corresponding to the decomposition of 30% DMF in **HOF-SXU-8** as discussed in the main text. During the second heating-cooling round, the sample was heated from room temperature to 160°C and then cooled to room temperature. During this round, there was no weight loss observed in the temperature range of 100-150°C. Weight loss continued only at temperatures above 153°C, which surpasses the boiling point of DMF. This indicates that after keeping at 110°C for two hours, the decomposition of DMF reached a maximum. Therefore it is concluded that the decomposition of DMF is accompanied by the melting of the **HOF-SXU-8** crystals, which leads to the gradual transformation of the crystals from an ordered state to a disordered state.

Fig. S13. The two rounds heating and cooling cycles TG curve of **HOF-SXU-8**.

According to the results of elemental analysis, it was deduced that there is a loss of approximately 30% DMF after glass formation (conclusion in the original main text). We reanalyzed the TG curve of **HOF-SXU-8** in Figure S4a and found that from 100-150°C, there was approximately a 7.1% weight loss, which corresponds to a 27% of total DMF in **HOF-SXU-8**, which is quite close to the 30% result obtained by heating the **HOF-SXU-8** under 110°C for 2 hours. From 150 to 310°C, there is a weight loss of about 28%, which corresponds to the total weight of all remaining DMF and H₂O molecules in **HOF-SXU-8**.

Fig. S4a. TG and heat flow of **HOF-SXU-8**.

The composition analysis was further carried out through liquid ^1H NMR spectroscopy of **HOF-SXU-8** and **HOF-SXU-8-g** (Figs. S11 and S12). The peaks at chemical shifts of 7.95, 2.89 and 2.73 ppm are assigned to H on the DMF molecule. By integrating the peak areas and comparing the change in the ratio of H on the naphthalene ring to the methyl H of DMF, it was found that the DMF molecule was reduced by about 29% after the transformation from **HOF-SXU-8** to **HOF-SXU-8-g**.

Fig. S11. ^1H NMR of **HOF-SXU-8** in $\text{DMSO-}d_6$ at room temperature.

Fig. S12. ¹H NMR of **HOF-SXU-8-g** in DMSO-*d*₆ at room temperature.

Combined the above analysis results, it reveals that: after heating between 110-150°C, about 30% of DMF in the **HOF-SXU-8** crystal is decomposed and **HOF-SXU-8** starts to melt, transforming into disordered state.

To clarify the importance of DMF in the glass transformation process, we have put **HOF-SXU-8-g** in saturated steam of DMF for 12 hours. The **HOF-SXU-8-g** was observed to re-crystallize (Fig. S9) back to **HOF-SXU-8** under heat treatment in DMF atmosphere, which is proved by the PXRD results (Fig. S10). However, no such re-crystallization was observed in the absence of DMF atmosphere, regardless of the same heat treatment. This result indicates that the occurrence of the glass transition behaviour is associated with the decomposition of the guest molecule DMF.

Fig. S9. **a** Diagram of **HOF-SXU-8-g** immersed in the DMF vapor atmosphere. **b** Photograph of **HOF-SXU-8-g** recrystallized after heating and cooling at 110°C in DMF atmosphere.

Fig. S10. Powder X-ray diffraction pattern of **HOF-SXU-8-g** before (purple) and after (blue) heating at 110°C and cooling in organic solvent DMF.

To observe complete melting peak, we have also re-conducted the DSC test by holding the temperature at 145°C for 2 hours as reviewer suggested (140-150°C). A clear melting peak appeared in the new DSC profile (Fig. 1) as reviewer expected. Therefore we have replaced the original DSC curve in Figure 1 with this new result for subsequent analyses. In the modified edition, we have also relabeled the figures to show the T_g in the the downscan in new Fig. 1b and Fig. S5.

Fig. 1. **a** DSC scans of crystalline **HOF-SXU-8** (Red curves represent the first heating and cooling cycle, blue curves represent the second cycles). **b** Enlarged figure of the circled part in Fig. 1a. **c** PXRD patterns of **HOF-SXU-8** and **HOF-SXU-8-g**. **d** FTIR spectra of **HOF-SXU-8** and **HOF-SXU-8-g**. **e** SEM image of the **HOF-SXU-8** pellet. **f** SEM image of the **HOF-SXU-8-g**. **g** Optical photographs of crystalline **HOF-SXU-8** after treated at different temperatures. The sample was put on the mark "X" to assess the variation in transparency.

Similarly, we have relabeled the temperature of the original DSC data obtained after kept at 110°C for 2 hours and rapid melt quenching (Fig. S6). The difference in heating temperatures resulted in the two samples displaying different T_g . This may be due to the difference in heat treatment temperature, which leads to the difference in the internal microstructure of the two samples, causing a subtle difference in T_g . Fig. S7 shows the DSC plot obtained after heating to 200°C and rapid melt quenching, no glass transition occurred due to the loss of larger amounts of DMF. The above experiments show that partial dissociation of guest DMF molecules is the key to vitrification.

Fig. S5. **a** DSC scans of crystalline **HOF-SXU-8** from -10 to 145 °C (red curves represent the first heating and cooling cycle, blue curves represent the second cycle). **b-c** Enlarged figures of the circled part in Fig. S5a.

Fig. S6. **a** DSC scans of crystalline HOF-SXU-8 from -10 to 110°C (red curves represent the first heating and cooling cycle, blue curves represent the second cycle). **b-c** Enlarged figures of the circled part in Fig. S6a.

Fig. S7. DSC scans of crystalline **HOF-SXU-8** from -10 to 200°C (red curves represent the first heating and cooling cycle, blue curves represent the second cycle).

Comment 2. The diameter of a 2032 steel coin cell is 20 mm. On the other hand, the samples used in the AC impedance measurements are 6 mm pellets which are not matched in size. When a sample is placed in a button cell, it is not possible to observe if the shape of the sample changes. How can the authors accurately determine sample area/height and subsequent conductivity after melting? The size and volume of such a soft compound changes greatly with temperature.

Response: Many thanks for your insightful question about the detail of ac impedance experiment. Placing the **HOF-SXU-8** pellet in a 2032 coin cell for in-situ testing aims to minimize the impact of environment humidity. During our experiments, we found that the melting temperature of the HOF is approximately 126°C (Fig. 1a). The actual heat treatment temperature was 110°C, which is lower than the melting point. According to the above discussion, we can know that the **HOF-SXU-8** crystals were slowly transformed into disordered phase due to the partial decomposition of DMF without obvious shape change. The mechanical properties of **HOF-SXU-8-g** was evaluated from viscosity and dynamic mechanical analyses (DMA). The storage modulus (G') dominated the loss modulus (G'') from 30 to 120°C (Fig. S15), suggesting that **HOF-SXU-8-g** has no fluidity even above the glass transition temperature. We have also conducted size measurements on samples before and after glass transition without placing them in the 2032 coin cell. The results indicate that there is no obvious changes in the thickness, and diameter of the pellet before and after the glass transformation. Based on the above facts, we sealed the HOF pellet samples in a 2032 steel coin cell in a glove box filled with ultra-pure Ar gas, with the

assumption that the dimensions of the HOF pellet do not change obviously before and after the glass transformation. This approach ensures a more precise assessment of the sample's properties while avoiding the environment humidity influence.

Fig. S15. DMA of **HOF-SXU-8-g** from -10 to 180°C (heating rate of 5°C min⁻¹). The storage (G') and loss (G'') moduli were marked as black and red, respectively.

Comment 3. Related to point 1 above, the results in Figure 2d alone are unsatisfactory considering the stability and durability of the sample. Cycle impedance measurements from low to high temperatures should be conducted several times. If they do not observe the same data set, it suggests that the measurements are not under equilibrium or that the sample shape is changing.

Response: Many thanks to you for providing valuable feedback on our work. In response to concerns regarding the stability and durability of the samples, we conducted additional experiments. Specifically, we have measured the impedance repeatedly from 30°C to 100°C. The results demonstrate that even after five rounds of proton conductivity tests, the proton conductivity can still be maintained at approximately $5 \times 10^{-2} \text{ S cm}^{-1}$ at 100°C. This suggests that the glassy **HOF-SXU-8-g** sample is under equilibrium and exhibits robust stability at 100°C. We have incorporated these new results into the updated discussion section of the manuscript. (Please see lines 181 to 184 on page 9 and the newly added Fig. S21)

Fig. S21. Proton conductivities of **HOF-SXU-8-g** measured across five heating and cooling cycles.

Comment 4. The mechanism of anhydrous proton conduction is still unclear: does the conductivity decrease when measured at temperatures above 100 degC? It is recommended to check the impedance at around 120 to 140 degC, which suggests (albeit indirectly) that the carriers of the conductivity are H^+ and H_3O^+ .

Response: Many thanks for your suggestions. We have re-conducted proton conduction experiments of **HOF-SXU-8-g** until 180°C according to your suggestion (Fig. S22). The results indicate that proton conductivity continues to increase up to 150°C. However, beyond 150°C, an obvious decline in proton conductivity is observed due to the simultaneous departure of guest molecules confined in **HOF-SXU-8-g** including DMF and H₂O, which is consistent with the TG results (Fig. S13). This suggests the crucial role of guest molecules in the proton conduction process, highlighting their importance in the system. The leaving of guest molecules makes the hydrogen-bonding network disrupted, and thus leads to the decrease in the proton conductivity. This phenomenon indirectly indicates that the carrier of proton conductivity is related to H^+ in H_3O^+ . The discussion section has been updated in the manuscript (please see lines 184 to 187 on page 10 and the newly added Fig. S22).

Fig. S22. Proton conductivities of **HOF-SXU-8-g** with temperature increasing.

Comment 5. Solid-state ^1H NMR has not been properly discussed. The authors should make a peak assignment of the HOF structure in the spectrum of Figure 3g before discussing the mobile proton peaks. It is not possible to describe the proton motion mainly from the chemical shifts. Peak sharpening should be observed at different temperatures or relaxation times should be measured.

Response: Thank you very much for your professional suggestion. We have investigated the dynamics of protons in **HOF-SXU-8-g** using variable-temperature solid-state ^1H MAS NMR as depicted in Fig. 3g. The peaks observed in the spectrum can be assigned as follows: those in the 6-10 ppm range correspond to protons on 1,5-NSA, while the peaks at 5 ppm are associated with protons on H_3O^+ , and those in the 0-4 ppm range are attributed to protons on DMF according to reference. The observed substantial narrowing and intensification of peaks within the temperature range of 30 to 100°C indicate a notable increase in the dynamics of all components. This phenomenon signifies an augmented mobility of H^+ within the structure as the temperature rises, contributing to an enhanced proton conductivity. We also re-tested the ^1H NMR of **HOF-SXU-8** as well as **HOF-SXU-8-g** and found that the peaks of these two are close to each other (Fig. S27), indicating that they are of the same composition. The DMF peaks in the **HOF-SXU-8-g** sample are weaker and broader, which may be related to the loss of DMF and exchange of active protons. We have modified our conclusions in the abstract and main text (please see lines 264 to 275 on page 13 and the newly added Fig. 3g and Fig. S27).

Fig. 3. **a** structure diagram of simulated **HOF-SXU-8-g** (cluster color: 1,5-NSA anion, red; DMF molecule, blue; H⁺, yellow; H₂O molecule, green). **b** Transient local structure of simulated **HOF-SXU-8-g** in MD simulation process (color code: C atom, pale blue; O atom, red; S atom, yellow; N atom, blue; H atom, white). **c** RDFs associated with 1,5-NSA anion and DMF molecules. **d** RDFs associated with H₂O molecules. **e** X-ray total scattering pattern of **HOF-SXU-8-g** at room temperature (blue) compared with that calculated from MD simulations (red). **f** Distribution of the number of hydrogen bonds over distance. **g** ¹H MAS solid-state NMR spectra of **HOF-SXU-8-g** at 30°C, 60°C and 100°C. **h** Self-diffusion coefficients of different components from **HOF-SXU-8-g** at different temperatures.

Fig. S27. ^1H MAS solid-state NMR spectra of **a** HOF-SXU-8 and **b** HOF-SXU-8-g at 30°C.

Reviewer 2:

Comments to the Author

Yang and co-authors reported in this manuscript vitrification as an effective strategy for the kinetically stable HOF material HOF-SXU-8 to eliminate HOF grain boundaries and obtain significantly enhanced proton conductivity without humidity. They have also investigated the transformation process and the structure after the glass transition through various experiments combined with molecular dynamics simulations. It is pointed out that protons play a connecting role between different clusters of the glassy structure and have a higher mobility than in the original HOF and thus much higher proton conductivity. This paper is well organized and contributes a meaningful scheme for the elimination of HOF grain boundary effect, which enables significant enhancement of proton conductivity. The proposed scheme has a number of advantages over state of the art and is expected to further guide the development of more super-protonic conductors. However, there are some problems which must be solved before it is considered for publication. If the following problems are well-addressed, I believe that the contribution of this paper is important for HOF applications. Therefore, I recommend publication after minor revisions.

Response: We thank the reviewer very much for the positive comment on our work as well as the valuable suggestions. Responses according to reviewer's suggestions are provided point-by-point as following:

Comment 1: Several important reviews about design rules and functionalities of stable HOFs are missing in the introduction.

Response: Thank you for your feedback. We have incorporated relevant literature regarding design rules and functionalities of stable HOFs into the introduction section of the manuscript (please see ref 5 and 22 on page 2).

Comment 2: The authors propose that HOF-SXU-8 is a kinetic product and HOF-SXU-9 is a thermodynamic product, can they give a mechanistic explanation as to why this transformation has occurred. Also, why can't HOF-SXU-9, whose main structure consists of the same 1,5-NSA molecules, undergo similar glass transition like HOF-SXU-8?

Response: Many thanks to the reviewer for the feedback on our work. In the process of crystal growth, two phases, kinetic and thermodynamic products, usually occur. The kinetic product **HOF-SXU-8** is formed during the growth process at a faster rate, while this structure has not reached the lowest free energy state. With the reaction time extending, the kinetic product **HOF-SXU-8** undergoes a phase transition to thermodynamic stable product **HOF-SXU-9**. The synthesis of **HOF-SXU-9** takes longer time because it involves processes such as rearrangement of atoms or molecules, surface energy adjustment, and so on. According to the principle of thermodynamics, **HOF-SXU-9** is in a lower energy state than **HOF-SXU-8**.

As for **HOF-SXU-8**, the glass transition occurs due to the removal or decomposition of DMF molecules in the system. Although the main structures of both **HOF-SXU-9** and **HOF-SXU-8** are composed of 1,5-NSA molecules, **HOF-SXU-9** does not involve the removal or decomposition of guest molecules during the heating treatment. From the TG result of **HOF-SXU-9** (Fig. S4), it can be seen that **HOF-SXU-9** does not undergo weight loss before 300°C, and what occurs after 300°C is the decomposition of the 1,5-NSA molecules. Therefore, it is concluded that **HOF-SXU-9** failed to undergo a glass transition due to the lack of free volume provided by the loss of guest molecules.

Comment 3: In Figure 1g, there are many visible bubbles after processed at 110°C, please explain this. Do these bubbles have any effect on the conductive properties of the proton? I am wondering whether these bubbles can be removed to form a better homogeneous phase of the glass or not.

Response: Thank you for your valuable feedback on our manuscript. **HOF-SXU-8** is transformed into glassy **HOF-SXU-8-g** after rapidly melt quenching. Due to the thermal decomposition of DMF under acidic conditions after heating at 110°C, the decomposition products will be evaporated at this temperature, which results in the presence of bubbles in the glass pellet. We have tried to slow down the heating rate, extend the heating time and many other means, but can not avoid the emergence of bubbles. In the presence of bubbles within the **HOF-SXU-8-g** glass pellet, the effective cross-sectional area for current conduction is reduced, thereby negatively affecting the glass pellet's electrical current conducting capacity. Consequently, the proton conductivity of the glass pellet is expected to improve upon the elimination of the bubbles.

Comment 4: The authors claim that their HOF-based glass exhibit superior proton conductivity. It is necessary to list the proton conductivities of HOF-based materials in a special table for comparison.

Response: Thank you for reviewer's suggestion. We have supplemented the table comparing the proton conductivity between **HOF-SXU-8-g** and other reported HOF-based materials in the literature (Table S6). (Please see Page S7)

Table S6. Summary of proton conductors based on HOFs.

	σ (S cm ⁻¹)	Conditions	Ea (eV)	Ref.
HOF-6a	1.9×10^{-6}	40°C (97%RH)	-	1
HOF-GS-10	7.5×10^{-3}	30°C (95%RH)	0.489	2
HOF-GS-11	1.8×10^{-2}	30°C (95%RH)	0.135	2
CPOS-1	1.0×10^{-2}	60°C (98%RH)	0.93	3
CPOS-2	2.2×10^{-2}	60°C (98%RH)	0.61	3
CPOS-3	3.3×10^{-4}	60°C (98%RH)	0.62	3
CPOS-4	7.4×10^{-4}	60°C (98%RH)	0.82	3
HOF-H ₃ L	6.91×10^{-5}	100°C (98%RH)	0.68	4
UPC-H1	5.0×10^{-3}	80°C (95%RH)	0.42	5
UPC-H2	2.6×10^{-3}	80°C (95%RH)	0.79	5
UPC-H3	4.3×10^{-2}	80°C (95%RH)	0.39	5
UPC-H5	1.71×10^{-3}	80°C (95%RH)	0.23	6
HOF 1	3.11×10^{-4}	70°C (98%RH)	0.71	7
HOF 2	4.32×10^{-4}	50°C (98%RH)	0.39	7
HDSD-1	7.50×10^{-3}	80°C (80%RH)	0.44	8
BPPA	5.14×10^{-2}	80°C (95%RH)	0.17	9
HOL-DMSO	4.42×10^{-2}	120°C (anhydrous)	0.42	10
HOF-SXU-1	6.32×10^{-3}	160°C (anhydrous)	0.16	11

HOF-SXU-8-g	5.62×10^{-2}	100°C (anhydrous)	0.23	This work
-------------	-----------------------	-------------------	------	-----------

Comment 5: In the reported research work on proton conductivity, cycling experiments on proton conductivity are usually included. This is important to measure whether a material can be practically applied. So does HOF-SXU-8-g maintain its performance after several rounds of warming and cooling experiments? Therefore, the authors should conduct additional experiments to verify this.

Response: Many thanks for reviewer's suggestions. We have measured the impedance repeatedly from 30°C to 100°C. The results demonstrate that even after five rounds of proton conductivity tests, the proton conductivity can still be maintained at approximately $5 \times 10^{-2} \text{ S cm}^{-1}$ at 100°C. This suggests that the glassy HOF-SXU-8-g sample is under equilibrium and exhibits robust stability at 100°C. We have incorporated these new results into the updated discussion section of the manuscript. (Please see lines 181 to 184 on page 9 and the newly added Fig. S21)

Fig. S21. Proton conductivities of HOF-SXU-8-g measured across five heating and cooling cycles.

Comment 6: Some of the sentences have grammatical mistakes and need to be adjusted accordingly, such as, in Page 9, The sentence "the system are summarized" contains a subject-verb agreement error. It should be "the system is summarized" to ensure agreement between the singular subject ("the system") and the singular verb ("is"). The manuscript needs careful polishing and particular attention to English grammar, spelling, and sentence structure.

Response : Many thanks to the reviewer for the kind reminder. We have addressed the specific issues pointed out by the reviewer, such as the subject-verb agreement error in the sentence on Page 11. The revised sentence now reads, "the system is summarized," ensuring proper agreement between the singular subject ("the system") and the singular verb ("is"). Additionally, we have conducted a thorough revision of the entire manuscript, paying meticulous attention to English grammar, spelling, and sentence structure with the changes highlighted in yellow. (Please see lines 207 on page 11)

Reviewer 3:

Comments to the Author

The authors here reported two kinds of HOFs constructed by 1,5-NSA, DMF and H₂O. Specifically, HOF-SXU-8 can be processed glassy state HOF-SXU-8-g by rapidly quenching approach. In contrast to HOF-SXU-8 (1.31×10^{-7} S cm⁻¹), HOF-SXU-8-g represented significantly enhanced anhydrous proton conductivity (5.62×10^{-2} S cm⁻¹), long-term stability, and temperature insensitivity. MAS NMR and MD simulation reveals that conductivity is mainly related to H⁺ on H₃O⁺. Generally, the result is novel and interesting, and I enjoy the reading of this manuscript. Therefore, the publication of this manuscript in Nat. Comm is recommended, but the following questions need to be addressed:

Response: We thank the reviewer very much for the positive comment on our work as well as the valuable suggestions. Responses according to reviewer's suggestions are provided point-by-point as following:

Comment 1: The authors claimed that T_g is 20.66°C (It might be more appropriate to describe it as 21°C). Does it mean HOF-SXU-8-g is a softer liquid-like state rather than glass state at 30 °C that most characterizations were conducted? The evidence in Figure 1b is doubtful. Please provide more evidences to affirm the state of HOF-SXU-8-g.

Response: Many thanks for reviewer's valuable comments. According to the reviewer's suggestion, we labeled the T_g of the sample studied in the main text as 21°C and placed it inside the support information part due to the following reason. We have re-conducted the DSC test to observe complete melting peak by holding the temperature at 145°C for 2 hours. A clear melting peak appeared in the new DSC profile (Fig. 1b). Therefore we have replaced the original DSC curve in Figure 1b.

To verify the state of **HOF-SXU-8-g**, the mechanical properties of **HOF-SXU-8-g** were evaluated from viscosity and dynamic mechanical analyses (DMA). The viscosity values (Fig. S14) of **HOF-SXU-8-g** at 30-120°C are lower than the viscosity at the Littleton softening point ($10^{6.6}$ Pa s), indicating that **HOF-SXU-8-g** exhibits some processing ability. Further increasing the temperature, the fluidity of **HOF-SXU-8-g** will rapidly become larger. The storage modulus (G') dominated the loss modulus (G'') from 30 to 120°C (Fig. S15), suggesting that **HOF-SXU-8-g** has

no fluidity even above the glass transition temperature (Chem. Sci., 2020, 11, 5175). We have incorporated these new results into the updated discussion section of the manuscript (please see lines 151 to 157 on page 8 and the newly added Figs. S14 and S15).

Fig. S14. Temperature-dependent viscosity of HOF-SXU-8-g.

Fig. S15. DMA of HOF-SXU-8-g from -10 to 180 °C (heating rate of 5 °C min⁻¹). The storage (G') and loss (G'') moduli were marked as black and red, respectively.

Comment 2: Page 5 Line 113, the authors claimed that T_m is 100.26 °C (It might be more appropriate to describe it as 100 °C) and HOF-SXU-8-g has fewer DMF molecules in structure than HOF-SXU-8. The boiling point of DMF is 153 °C and there are hydrogen bonding between DMF and H₂O in structure. Furthermore, based on the results from MD calculation, there are almost no interactions between water and DMF/1,5-NSA anion molecules in HOF-SXU-8-g. Why DMF can be removed at 110 °C rather than H₂O? Apart from the evidence from elemental analysis, additional experiments, such as ¹H NMR, should be added to support authors' conclusion.

Response: Many thanks to the reviewer for the professional suggestions. Due to two hours heat treatment at 110°C, it can not display the complete melting behavior of **HOF-SXU-8**. We have re-conducted the DSC test by holding the temperature at 145°C for 2 hours and a clear melting peak appeared in the new DSC profile (new Fig. 1a). The T_m was re-calculated to be integer 126°C according to the reviewer's suggestion. About the question why it is not H₂O removed at 110°C raised by the reviewer, we speculate that H₂O molecules in **HOF-SXU-8** structure are in the form of H₃O⁺, and the H₃O⁺ molecules in the structure are bounded to 1,5-NSA anions and DMF molecules through complex hydrogen bonding. Therefore, the H₂O is hard to remove even above 100°C. However, DMF is easy to decompose under conditions of strong acidity (-SO₃H in **HOF-SXU-8** provide) and elevated temperatures (Tetrahedron **2009**, 65(40), 8313-8323). After DMF decomposition, these H₃O⁺ molecules recombined to form stable clusters. These H₃O⁺ clusters have strong interactions with the 1,5-NSA anions according to MD simulation results and are immobilized within the glass pellet. As a result, the water molecules are well retained in the system.

As discussed in the main text, we have demonstrated through a series of studies that the weight loss at around 100°C is not directly originated from the evaporation of DMF, but from the thermal decomposition of partial DMF under strong acidic conditions. The low boiling point products produced by the decomposition can be easily evaporated at temperatures above 100°C. The decomposition of DMF is accompanied by melting of the **HOF-SXU-8** crystals, which results in the gradual transformation of the crystals from an ordered state to a disordered state. To determine the composition of the system we have added some experiments and specific experimental details are listed below:

Firstly, according to the reviewer's suggestion, the analysis was carried out by liquid ¹H NMR spectroscopy and the peaks at chemical shifts 7.95, 2.89 and 2.73 ppm corresponded to H on the DMF molecule. By integrating the peaks and comparing the ration of H on naphthalene to H on DMF, it was found that there was a reduction of about 29% of the DMF molecule after the transformation of **HOF-SXU-8** to **HOF-SXU-8-g** at 110°C (Fig. S11 and S12).

Fig. S11. ¹H NMR of HOF-SXU-8 in DMSO-*d*₆ at room temperature.

Fig. S12. ¹H NMR of HOF-SXU-8-g in DMSO-*d*₆ at room temperature.

Secondly, we re-analyzed the TG of **HOF-SXU-8** (Fig. S4), which is calculated to lose about 7.1% of weight in the temperature range of 100 to 153°C. The calculation reveals that about 27% of DMF is lost, which is consistent with the conclusion obtained from our elemental analysis and liquid ^1H NMR spectroscopy. In the range of 153-315°C, the weight loss is about 28%, which corresponds exactly to the proportion of H_2O molecules and the remaining DMF molecules in the structure. We have performed two rounds of heating-cooling cycles on **HOF-SXU-8** to measure its weight loss behavior (Fig. S13). First, **HOF-SXU-8** crystals were heated to 110°C, kept at 110°C for 2 hours, and then cooled from 110°C to room temperature. This process resulted in about 7.3% weight loss of the sample, roughly corresponding to 30% of DMF in **HOF-SXU-8**. The cooled sample was further heated from room temperature to 160°C and then cooled to room temperature. There was no weight loss between in the temperature range of 100-153°C during this process. The above results indicate that after keeping **HOF-SXU-8** at 110°C for two hours, the decomposition of DMF reached a maximum.

Fig. S13. The two rounds heating and cooling cycles TG curve of **HOF-SXU-8**.

Thirdly, as discussed in the original manuscript, elemental analysis showed that the molecular formulas were $(1,5\text{-NSA})_{1.04} (\text{H}_2\text{O})_{2.25} (\text{DMF})_{2.2}$ for **HOF-SXU-8** and $(1,5\text{-NSA}) (\text{H}_2\text{O})_{2.26} (\text{DMF})_{1.56}$ for **HOF-SXU-8-g** (Table S5), with the difference related to the loss of approximately 30% DMF after glass formation.

Fourthly, **HOF-SXU-8-g** was observed to re-crystallize (Fig. S9) back to **HOF-SXU-8** under heat treatment in pure DMF atmosphere, which is proved by the consistent PXRD results (Fig. S10). However, no such re-crystallization was observed in the absence of DMF atmosphere, regardless of the same heat treatment temperature.

This result indicates that the occurrence of the glass transition behaviour is associated with the loss of the guest molecule DMF.

In conclusion, these above results are consistent, where the glassy behaviour occurs as a result of the decomposition of about 30% DMF.

Comment 3: Since this material is not crystalline, how the authors can determine the molecular formulas? Please elucidate the detail procedure on establishing the molecular formulas of HOF-SXU-8-g . Furthermore, HOF-SXU-8-g has obvious difference between calculated and experimental elemental analysis for C. Please provide an explanation.

Response: Many thanks for the reviewer's reminder. The molecular formulas for non-crystalline **HOF-SXU-8-g** was determined through the following detailed procedure: Initially, the composition of **HOF-SXU-8** was used as a starting formula. Then we manually adjusted the proportions of those components (1,5-NSA, H₃O⁺, DMF) while monitoring the changes in elemental content using the analysis function in chemdraw software. This iterative process continued until the composition closely matched the results obtained from experimental measurements. Based on the results of the above tests, we determined the molecular formula of **HOF-SXU-8-g** to be (1,5-NSA) (H₃O)_{2.26} (DMF)_{1.56}.

The final experimental results were in good agreement with above supplementary experiments such as TG analysis and ¹H NMR results (see response 2 for specific analyses). However, there was an 0.8% difference between the calculated and experimental elemental analysis for C. We suggest that this discrepancy could be attributed to the presence of minor decomposition products of DMF within the bubbles of HOF glass pellet, potentially causing some deviation in the carbon content.

Comment 4: Please provide the table for comparing the proton conductivity between HOF-SXU-8-g and other reported materials.

Response: Many thanks to the reviewer for the useful comments on our work. We have supplemented the table comparing the proton conductivity between **HOF-SXU-8-g** and other reported materials in the literature (Table S6). (Please see Page S7)

Table S6 Summary of proton conductors based on HOFs.

	σ (S cm ⁻¹)	Conditions	Ea (eV)	Ref.
HOF-6a	1.9×10^{-6}	40°C (97%RH)	-	1
HOF-GS-10	7.5×10^{-3}	30°C (95%RH)	0.489	2
HOF-GS-11	1.8×10^{-2}	30°C (95%RH)	0.135	2
CPOS-1	1.0×10^{-2}	60°C (98%RH)	0.93	3
CPOS-2	2.2×10^{-2}	60°C (98%RH)	0.61	3
CPOS-3	3.3×10^{-4}	60°C (98%RH)	0.62	3
CPOS-4	7.4×10^{-4}	60°C (98%RH)	0.82	3
HOF-H ₃ L	6.91×10^{-5}	100°C (98%RH)	0.68	4
UPC-H1	5.0×10^{-3}	80°C (95%RH)	0.42	5
UPC-H2	2.6×10^{-3}	80°C (95%RH)	0.79	5
UPC-H3	4.3×10^{-2}	80°C (95%RH)	0.39	5
UPC-H5	1.71×10^{-3}	80°C (95%RH)	0.23	6
HOF 1	3.11×10^{-4}	70°C (98%RH)	0.71	7
HOF 2	4.32×10^{-4}	50°C (98%RH)	0.39	7
HDSD-1	7.50×10^{-3}	80°C (80%RH)	0.44	8
BPPA	5.14×10^{-2}	80°C (95%RH)	0.17	9
HOL-DMSO	4.42×10^{-2}	120°C (anhydrous)	0.42	10
HOF-SXU-1	6.32×10^{-3}	160°C (anhydrous)	0.16	11
HOF-SXU-8-g	5.62×10^{-2}	100°C (anhydrous)	0.23	This work

Comment 5: Variable-temperature solid-state NMR should be added to provide insight into structural dynamics.

Response: Many thanks for reviewer's suggestion. We have investigated the dynamics of protons in **HOF-SXU-8-g** using variable-temperature solid-state ¹H MAS NMR as depicted in Fig 3g. The peaks observed in the spectrum can be assigned as follows: those in the 6-10 ppm range correspond to protons on 1,5-NSA, while the peaks at 5 ppm are associated with protons on H₃O⁺, and those in the 0-4 ppm range are

attributed to protons on DMF according to reference. The observed substantial narrowing and intensification of peaks within the temperature range of 30 to 100°C indicate a notable increase in the dynamics of all components. This phenomenon signifies an augmented mobility of H⁺ within the structure as the temperature rises, contributing to an enhanced proton conductivity. (Please see lines 264 to 275 on page 13 and the newly added Fig. 3g)

Fig. 3. **a** structure diagram of simulated **HOF-SXU-8-g** (cluster color: 1,5-NSA anion, red; DMF molecule, blue; H⁺, yellow; H₂O molecule, green). **b** transient local structure of simulated **HOF-SXU-8-g** in MD simulation process (color code: C atom, pale blue; O atom, red; S atom, yellow; N atom, blue; H atom, white). **c** RDFs associated with 1,5-NSA anion and DMF molecules. **d** RDFs associated with H₂O molecules. **e** X-ray total scattering pattern of **HOF-SXU-8-g** at room temperature (blue) compared with that calculated from MD simulations (red). **f** Distribution of the number of hydrogen bonds over distance. **g** ¹H MAS solid-state NMR spectra of **HOF-SXU-8-g** at 30°C, 60°C and 100°C. **h** Self-diffusion coefficients of different components from **HOF-SXU-8-g** at different temperatures.

Comment 6: Add the fitting of experimental data from pair distribution function (PDF).

Response: Many thanks for reviewer's suggestions. The X-ray total scattering data obtained from experimental data were corrected, normalized, and Fourier transformed to obtain a reduced radial distribution function (Figure S25). Below 8 Å, the experimental PDF pattern shows sharp peaks, suggesting that the **HOF-SXU-8-g** glass might be ordered at short distances. Similar sharp peak features are also observed in the PDF of ZIF glasses (Chem. Commun. **2019**, 55, 2521). While beyond 8 Å, the experimental PDF exhibits much weaker peak, indicating the disorder in long-range. Besides, we have validated our simulation results by obtaining the X-ray total scattering pattern of amorphous **HOF-SXU-8-g** and comparing it with the simulated X-ray total scattering pattern based on the MD simulation model (Fig. 3e). This has the same effect as the method of comparing the PDF data obtained from experimental data and MD simulation model (J. Appl. Crystallogr. **2003**, 36, 1342; J. Appl. Crystallogr. **2013**, 46, 560; Acta Crystallogr., Sect. A **2015**, 71, 562). (Please see lines 215 to 220 on page 11 and the newly added Fig. S25)

Fig. S25. Experimental pair distribution function of **HOF-SXU-8-g**.

Comment 7: I can find an obvious difference between experimental and calculated proton conductivity. Please provide an explanation.

Response: Many thanks to the reviewer for the feedback on our work. The deviation of the proton conductivity between experimental and calculated proton conductivity falls in the same order of magnitude, which is acceptable (J. Mol. Liq. **2013**, 187, 238-245; J. Chem. Phys. **2020**, 153, 024116). The reason is explained as follows:

Firstly, molecular dynamics simulations usually use potential function models to describe the interactions between atoms or molecules. The potential function model is

always not accurate enough for accurately describing the actual physical mechanism of the proton transport process. The chosen parameters of these potential function models may also affect the accuracy of simulation results. Therefore the calculated proton conductivity may deviate from the experimental results.

Secondly, the conditions such as temperature and pressure used in the simulation can not be completely consistent with the experimental conditions. Differences in these conditions may cause the simulation results to deviate from the experimental results.

Thirdly, the size of the system considered in the molecular dynamics simulation can affect the simulation results. The size of the simulated system does not accurately reflect the real situation, which could also lead to the calculated proton conductivity biased from experimental result. In addition to this: molecular dynamics simulations are usually based on classical mechanics, which may not capture certain quantum mechanical effects or kinetic effects that are important for describing the proton transport behaviour.

In summary, we believe that the appearance of this deviation is acceptable (please see lines 260 to 262 on page 13).

Comment 8: Page 6 Figure 1a, clarify why there are four lines for two samples?

Response: Thank you for the reviewer's reminder. Since the observation of the glass transition temperature requires the monitoring of the thermal behavior of one material during the heating and cooling cycles. We have measured **HOF-SXU-8** for two DSC cycle experiments, so there are four lines appeared in Figure 1a. The red curve represents the first heating and cooling cycle and the blue curve represents the second heating and cooling cycle (please see lines 115 to 117 on page 6 and Fig. 1a).

Fig. 1. **a** DSC scans of crystalline **HOF-SXU-8** (Red curves represent the first heating and cooling cycle, blue curves represent the second cycles). **b** Enlarged figure of the circled part in Fig. 1a. **c** PXRD patterns of **HOF-SXU-8** and **HOF-SXU-8-g**. **d** FTIR spectra of **HOF-SXU-8** and **HOF-SXU-8-g**. **e** SEM image of the **HOF-SXU-8** pellet. **f** SEM image of the **HOF-SXU-8-g**. **g** Optical photographs of crystalline **HOF-SXU-8** after treated at different temperatures. The sample was put on the mark "X" to assess the variation in transparency.

Comment 9: Compared to T_g of ZIF glass (>400°C), why does HOF have a much lower T_g?

Response: Many thanks for reviewer's question. Metal-organic framework materials and hydrogen-bonded organic framework materials are two different types of materials, which have different structures and properties, and therefore different glass transition temperatures. MOF is a porous crystalline material consisting of metal ions and organic ligands, and its structure exhibits certain rigidity. The glass transition temperature of MOF is usually higher due to the high crystallinity and stability of MOF, which require higher temperature to cause them to lose its orderliness. The glass transition temperature of HOF is usually lower because the intermolecular hydrogen bonding in HOF is weaker than that in MOF and the molecules are more free to move, which makes HOF easy to lose its orderliness at lower temperature. Overall, the difference in glass transition temperatures between MOFs and HOFs mainly stems from the differences in the strength of hydrogen bonding with respect to metal-ligand bonding. In our work, the structure of **HOF-SXU-8** is mainly assembled through hydrogen bonding, so its glass transition temperature T_g is much lower than that of ZIF glass (>400°C).

Comment 10: Page 8 Line 153, detailed simulation description is in "methods section" in Manuscript, not in SI.

Response: Thank you for your reminder. We've modified the SI to "methods section", which has been highlighted in yellow. (Please see Line 192 on Page 10)

REVIEWER COMMENTS

Reviewer #1 (Remarks to the Author):

I understand the reason for bubbles is due to heating. The term "melting" phenomenon is applicable when a compound/substance that does not change composition exhibits a solid-to-liquid transition. The liquidation accompanied by a large amount of DMF release cannot be regarded as a "melting".

For example, Figure 1C shows "HOF-SXU-8" and "HOF-SXU-8-g," the former containing a large amount of DMF and the latter not. The compositions of these two states are different, so these name are not appropriate. In addition, the photograph of HOF-SXU-8-g with many bubbles (previously pointed out in Scheme 1) should be replaced with a photograph of the bubble-free glass texture by completely removing DMF.

DMA measurements were performed in Figure S15. Is this an upscan process or a downscan? It should be clarified. They concluded that there is no fluidity because G' is greater than G'' in the temperature regime, but that discussion is insufficient. For example, it is strange that there is no change at all in G' and G'' near the glass transition point.

On page 12 of the rebuttal letter, the authors write, "due to the simultaneous departure of guest molecules confined in

HOF-SXU-8-g including DMF and H₂O". This clearly indicates that the internal water contributes to proton conduction. On the other hand, Line 281 in the main text says "anhydrous," which creates a contradiction.

Figure 3g shows a solid-state ¹H NMR measurement at variable temperature. The authors state that the line width decreases as the temperature increases, but this is not accurate. The spectra show little change in FWHM, and the argument needs to be revised. If we aim to study the mobility of the proton by SSNMR, relaxation time measurements would be necessary in this case.

Overall, the authors have performed additional experiments and tried to improve the quality of the paper's argument, but it is still not convincing enough for the observed phenomena. Interpretation of each state in the sample is confusing from the viewpoint of composition, glass, supercooled liquid, with/without guest solvents.

Reviewer #2 (Remarks to the Author):

It is always gratifying when an author chooses to take on board referees' comments, and uses them to improve the quality of the paper. The clarification of the experimental details makes this work more scientifically sound and significantly enhance the importance of this paper, and I am now happy to recommend acceptance.

Reviewer #3 (Remarks to the Author):

All my comments have been addressed. I now support the publication of the manuscript.

Response to Reviewer 1 Comments

Reviewer 1:

Response: We appreciate your attention to our work and we extend our sincere thanks for your valuable feedback. All of your suggestions are very important and have important guiding significance for my paper writing and research work. According to the suggestions, we have thoroughly reviewed the results and discussion section and addressed the major concerns you have raised point by point. Your constructive suggestion is highly valued, and we have made the necessary revisions to ensure the quality of our manuscript. Thank you again for your time and insights. Corresponding revisions and point-to-point responses are listed as following:

Comments to the Author

Comment 1: I understand the reason for bubbles is due to heating. The term "melting" phenomenon is applicable when a compound/substance that does not change composition exhibits a solid-to-liquid transition. The liquidation accompanied by a large amount of DMF release cannot be regarded as a "melting".

Response: Many thanks for your comments and suggestions. We acknowledge the reviewer's point regarding the term "melting" and its applicability to our study. The phenomenon observed, characterized by the release of a significant amount of DMF during heating, does not align with the conventional definition of "melting" for a compound maintaining its composition during a solid-to-liquid transition. We have avoid the use of "melting" in the main text to more accurately describe the observed process. Thank you for guiding us in improving the clarity and accuracy of our work.

Comment 2: For example, Figure 1C shows "HOF-SXU-8" and "HOF-SXU-8-g," the former containing a large amount of DMF and the latter not. The compositions of these two states are different, so these name are not appropriate. In addition, the photograph of HOF-SXU-8-g with many bubbles (previously pointed out in Scheme 1) should be replaced with a photograph of the bubble-free glass texture by completely removing DMF.

Response: Many thanks for your valuable feedback. We acknowledge the reviewer's point regarding the names "HOF-SXU-8" and "HOF-SXU-8-g" and have made the

appropriate changes. To better distinguish between the two states, we have revised "HOF-SXU-8-g" to "HOF-g" in the manuscript. Additionally, in Scheme 1, we have replaced the image with a bubble-free representation to accurately reflect the desired state. We made several attempts to create a bubble-free "HOF-g" sample but have been unsuccessful thus far. Therefore, we have substituted the image with a bubble-free model to convey the intended representation. We greatly appreciate the reviewer's attention to detail, and these adjustments have been made accordingly. (please see lines 58 to 60 on page 3 and the newly added Scheme 1).

Scheme 1. Schematic representation of the synthesis of **HOF-SXU-8**, **HOF-SXU-9** and the vitrification transformation process to **HOF-g**. (Color code: C atom, gray; O atom, red; S atom, yellow; N atom, blue; H atom, white).

Comment 3: DMA measurements were performed in Figure S15. Is this an upscan process or a downscan? It should be clarified. They concluded that there is no fluidity because G' is greater than G'' in the temperature regime, but that discussion is insufficient. For example, it is strange that there is no change at all in G' and G'' near the glass transition point.

Response: We appreciate your insightful feedback and have provided additional clarification based on your suggestions. The DMA measurement performed in Figure S15 is an upscan process. In Figure S15, the storage modulus (G') was observed to dominate over the loss modulus (G'') from 30 to 120°C, indicating the lack of fluidity in HOF glass even above the glass transition temperature. The immediate reduction of G' above 120°C signifies the softening of HOF-g, as previously discussed in reported publications (Chem. Sci., 2021, 12, 5818-5824; Chem. Sci., 2020, 11, 5175-5181).

As for the lack of observable changes in G' and G'' near the glass transition point, this can be attributed to the relatively stable molecular structure of the material below and above the glass transition temperature. The material does not undergo significant structural changes in response to temperature variations, resulting in minimal alterations in both storage modulus (G') and loss modulus (G''). Furthermore, the storage modulus (G') and loss modulus (G'') of the sample exhibit low sensitivity to temperature changes, which contributes to the observed stability near the glass transition point.

Our analysis of the storage (G') and loss modulus (G'') curves around 150°C suggested that HOF glass resides between a high elastic state and a viscous flow state near this critical temperature point. Upon increasing the temperature to 160°C , G' drastically decreased from approximately 10^6 Pa to about 10^2 Pa due to structural damage to the HOF glass (Chem. Mater. 2018, 30, 11, 3752-3758; Polym. Chem., 2023, 14, 1536).

Fig. S15. DMA of HOF-glass from -10 to 180°C (heating rate of $5^{\circ}\text{C min}^{-1}$). The storage (G') and loss (G'') moduli were marked as black and red, respectively.

Comment 4: On page 12 of the rebuttal letter, the authors write, "due to the simultaneous departure of guest molecules confined in HOF-SXU-8-g including DMF and H_2O ". This clearly indicates that the internal water contributes to proton conduction. On the other hand, Line 281 in the main text says "anhydrous," which creates a contradiction.

Response: We greatly appreciate your attention to detail, which has helped us improve the clarity and accuracy of our work. As previously mentioned, the structure of HOF-SXU-8-g (now denoted as HOF-g) contains both DMF and H_2O , with H_2O

playing a significant role in proton conduction. Regarding the term "anhydrous" mentioned on line 281 of the manuscript, we would like to clarify that it refers to the proton conductivity under conditions without humidity. We recognize that this may have caused confusion, given the presence of H₂O in the structure. Therefore, we have updated the manuscript to replace "anhydrous" with "without humidity" to better reflect the context. This modification can be found in lines 280 to 282 on page 14 of the revised manuscript.

Comment 5: Figure 3g shows a solid-state ¹H NMR measurement at variable temperature. The authors state that the line width decreases as the temperature increases, but this is not accurate. The spectra show little change in FWHM, and the argument needs to be revised. If we aim to study the mobility of the proton by SSNMR, relaxation time measurements would be necessary in this case.

Response: Thank you for your valuable suggestions. Your insights have been incredibly helpful for our study. In the variable temperature solid-state ¹H NMR analysis, the observed overlapping of peaks indeed results in minimal changes in the full width at half maximum (FWHM) with increasing temperature. Therefore, it cannot solely explain the enhanced mobility of protons in the system as temperature rises. In response to your recommendation, we have included the measurement of T_{2H} relaxation times for HOF-g at different temperatures (refer to Figure S28). The T_{2H} values for HOF-g at 30, 60, and 100°C were found to be 0.4318, 0.4898, and 0.4947 ms, respectively, indicating a gradual increase in relaxation time with rising temperature. These additional solid-state NMR analyses, particularly the T_{2H} relaxation time measurements, further support our findings. They suggest that the proton mobility within the HOF-g structure is indeed enhanced as the temperature increases. This enhanced mobility contributes to the improved proton conductivity properties observed. We have incorporated this discussion and the relevant data into the revised manuscript for clarity and completeness. Please refer to lines 283 to 293 on page 14 and the newly added Figure S28 for more details.

Fig. S28. T_{2H} of ^1H MAS solid-state NMR signals of **HOF-g** with temperature increasing.

REVIEWER COMMENTS

Reviewer #1 (Remarks to the Author):

If you are going to put an imaginary picture of a glass in Scheme 1 that is not the actual one you got, I recommend you to back to the real photo of the actual sample in the previous version although it contains bubbles.

The relaxation time section of solid-state NMR: It is dangerous because the author put it without understanding the definition and discussion of relaxation time. First of all, the term "T2H" is wrong, and there are three relaxation times: T1, T2, and T1rho. The author is probably referring to T2 here, but for the mobility study, T1 seems to be a more appropriate measurement. Also, you added Figure S28, but there is no discussion of what part of the HOF structure you are investigating, and why this relaxation time measurement shows an increase in mobility. It would be better not to include the relaxation time measurement results if they are not able to discuss the relationship between relaxation time and structural dynamics.

Response to Reviewer 1 Comments

Reviewer 1:

Comment 1: If you are going to put an imaginary picture of a glass in Scheme 1 that is not the actual one you got, I recommend you to back to the real photo of the actual sample in the previous version although it contains bubbles.

Response: Many thanks for your comments and suggestions. Certainly, we acknowledge the reviewer's suggestion and we have replaced the imaginary picture with the original image of the glass sample with bubbles in the previous version. (please see lines 58 to 60 on page 3 and the newly added Scheme 1).

Scheme 1. Schematic representation of the synthesis of **HOF-SXU-8**, **HOF-SXU-9** and the vitrification transformation process to **HOF-g**. (Color code: C atom, gray; O atom, red; S atom, yellow; N atom, blue; H atom, white).

Comment 2: The relaxation time section of solid-state NMR: It is dangerous because the author put it without understanding the definition and discussion of relaxation time. First of all, the term " T_{2H} " is wrong, and there are three relaxation times: T_1 , T_2 , and $T_{1\rho}$. The author is probably referring to T_2 here, but for the mobility study, T_1 seems to be a more appropriate measurement. Also, you added Figure S28, but there is no discussion of what part of the HOF structure you are investigating, and why this relaxation time measurement shows an increase in mobility. It would be better not to include the relaxation time measurement results if they are not able to discuss the relationship between relaxation time and structural dynamics.

Response: Thank you for your valuable feedback on our manuscript regarding the relaxation time section in solid-state NMR. We appreciate the opportunity to address your concerns and clarify our approach. Regarding the terminology " T_{2H} ," we understand your point, and we acknowledge that it may not align with conventional nomenclature. However, we would like to emphasize that this term was used in accordance with previous literature reference, where it was employed in a similar context (J. Am. Chem. Soc. 2023, 145, 9808-9814). In another reference by the same research group, the relationship between relaxation time and structural dynamics was discussed using relaxation time T_2 (Chem. Sci., 2020, 11, 5175-5181), although we recognize that for the study of mobility, the T_1 measurement may indeed be more appropriate.

Furthermore, we apologize for the lack of clarity regarding the specific aspects of the HOF structure under investigation in Figure S28. The relaxation time measurements were meant to highlight an increase in mobility within the material, as observed in previous studies. However, we acknowledge that without a thorough discussion of the relationship between relaxation time and structural dynamics, the inclusion of these results may not add significant value to the manuscript. Our primary focus in this study is on the glass transition of HOFs, the characterization of their structure, and the investigation of proton conductivity properties within this framework. As the discussion on the local structural proton dynamics does not directly contribute to the main findings of the paper, we have decided to heed your suggestion and remove the relevant sections on nuclear magnetic resonance (NMR) testing. (please see lines 277 to 293 on page 14).